# Dough Rheological Behavior and Microstructure Characterization of Composite Dough with Wheat and Tomato Seed Flours

**DOI:** 10.3390/foods8120626

**Published:** 2019-11-30

**Authors:** Silvia Mironeasa, Georgiana Gabriela Codină

**Affiliations:** Faculty of Food Engineering, Stefan cel Mare University of Suceava, 720229 Suceava, Romania; silviam@fia.usv.ro

**Keywords:** tomato seed flour, wheat flour, dough rheology, microstructure

## Abstract

The rheological and microstructural aspects of the dough samples prepared from wheat flour and different levels of tomato seed flour (TSF) were investigated by rheology methods through the Mixolab device, dynamic rheology and epifluorescence light microscopy (EFLM). The Mixolab results indicated that replacing wheat flour with TSF increased dough development time, stability, and viscosity during the initial heating-cooling cycle and decreased alpha amylase activity. The dynamic rheological data showed that the storage modulus *G’* and loss modulus *G”* increased with the level of TSF addition. Creep-recovery tests of the samples indicated that dough elastic recovery was in a high percentage after stress removal for all the samples in which TSF was incorporated in wheat flour. Using EFLM all the samples seemed homogeneous showing a compact dough matrix structure. The parameters measured with Mixolab during mixing were in agreement with the dynamic rheological data and in accordance with the EFLM structure images. These results are useful for bakery producers in order to develop new products in which tomato seed flour may be incorporated especially for wheat flours of a good quality for bread making and high wet gluten content. The addition of TSF may have a strength effect on the dough system and will increase the nutritional value of the bakery products.

## 1. Introduction

Tomato (*Lycopersicon esculentum*) is one of the most consumed vegetables, as raw (fresh), cooked, in food preparations and as processed products such as tomato juice, puree, paste, ketchup, sauce and canned tomato. Due to its nutritional and bioactive components, the foods products in which tomato is incorporated represents a valuable source of minerals, vitamins and antioxidants, namely lycopene, which varies according to the tomato variety, maturity and agro-environmental factors during growth [1,2]. Post-harvest storage conditions [3,4] and processing technology [5] also influence the tomatoes and tomato-based food products. Dietary intake of tomatoes and processed tomato products has been associated with a decrease in susceptibility to chronic diseases such as various type of cancers and cardiovascular diseases [6]. The medical or health benefits of tomato intake are attributed to the natural antioxidant compounds present in tomato such as ascorbic acid, carotenoids (mainly lycopene) and phenolic compounds which play a crucial role in the health protection mechanisms by scavenging free radicals [7].

Processing of tomatoes leads to a high amount of by-products, about 3–7% of the tomato weight [8,9] that generates serious environmental problems for the industry concerned due to the disposal of the organic material. Tomato pomace, the major part of the by-products, includes mainly seeds and peels in various proportions and a small amount of pulp [10]. Some amounts of by-products are used in animal feed or as soil fertilizers [11]. However, today there is an increasing trend of using this by-product as a source of functional components in different products knowing that the tomato pomace is a rich source of valuable compounds which can be recovered and used in food, pharmaceutical and cosmetic industries as natural ingredients [12,13]. In the chemical composition of tomato pomace high level of total dietary fibers, proteins, fats and medium amounts of ash are found [14].

The peel and seed by-products of tomato pomace represents approximately 20–50 g⋅kg^−1^ of the initial weight of tomatoes and can be used individually or combined [14]. Regarding the combined utilization, the literature reports that, due to the significant amounts of bioactive phytochemicals from tomato pomace, it can be used as natural antioxidants for the formulation of functional foods, or as additives in food systems to extend their shelf-life [15,16,17]. Other studies showed that tomato pomace can be considered a good source of some macroelements, such as potassium, manganese and calcium and microelements, i.e., copper and zinc, which are cofactors of the antioxidant enzymes [15]. As individual part of by products, many studies [16,17,18,19] highlighted the possibility of using the tomato peels as a source of carotenoids and antioxidants to enrich various foods products or to produce new functional ones. The other part of tomato by-product, tomato seeds, account for approximately 10% of the fruit and 60% of the total by-product and contain a high amount for protein, fat and mineral elements such as potassium, calcium, iron, manganese, zinc and copper [14,20]. Tomato seed proteins present adequate properties to form a good emulsion in a food system [21]. An improved protein quality for bread supplemented with 10% seed meal was reported by Sogi et al. [22]. According to Kramer and Kwee [23] the nutritive value of tomato seed protein is lower than of casein but equivalent or higher than of other plant proteins. The net protein retention (NPR) of whole tomato seed meal, defatted tomato seed meal and tomato seed protein concentrate was reported as 2.65%, 2.52% and 2.51%, respectively which were lower but comparable with those obtained for casein, 2.91% (dry basis) [24].

High levels of essential amino-acids present in tomato seeds showed that this by-product presents high quality proteins, with a high amount of lysine (3.4–5.9%) [25,26,27]. The amount of lysine in tomato seed is approximately 13% higher than the amount found in the soy protein [28]. Therefore, the tomato seed may be recommended to fortify various low-lysine food products that are deficient in this amino acid. One of these products are the bakery ones of which the base raw material is wheat flour which contains low amounts of lysine. In addition, compared to other seed sources, no anti-nutritional constituents have been reported in tomato seeds [29], a fact that make them a better source of proteins compared to other non-conventional sources.

The use of tomato seeds in order to improve the quality of food products were less studied. These studies were focused especially on the use of the dried pomace [30,31] or tomato peels in food products [32,33]. However, the use of tomato seed in bread making has been previously reported in a few studies [34,35,36] which were especially referring to the tomato seed addition effect on bread quality. To our knowledge, the information on the effect of partial substitution of wheat flour with whole tomato seed flour on dough rheological properties are missing. No studies have been made on the effect of wheat flour substitution with tomato seed flour at the levels of 5%, 10%, 15% and 20% on dough rheological properties and its microstructure. The use of this kind of methodology, which investigates both empirical and fundamental rheological properties on this subject, is not frequent. Also, the investigation of dough microstructure through a modern device namely epifluorescence light microscopy (EFLM) helped us to better understand the wheat flour dough behavior with different levels of tomato seed flour (TSF) addition during the bread making technological process. The objective of this study was: (1) to evaluate the physicochemical characteristics for the wheat–tomato seed composite flours; (2) to investigate the changes that occur in the dough formed by the wheat–tomato seed composite flours during mixing as well as the quality of starch and protein from the dough system by using the Mixolab device; (3) to analyze the effect of tomato seed flour addition (TSF) on the fundamental dough rheological properties by using a rheometer on which oscillatory frequency test and creep recovery test was performed; (4) to analyze the effect of TSF on dough microstructure, further analyzed through epifluorescence light microscopy (EFLM).

## 2. Materials and Methods

### 2.1. Flour Samples

One commercial refined wheat flour type 650 was used. The wheat flour was provided by S.C. Mopan S.A. Company (Suceava, Romania), a local milling company. Tomato seed flour was collected from tomatoes (*Solanum lycopersicum*) cultivated in Suceava County, Romania. The tomato seeds were separated and cleaned after our own developed procedure from the tomato by-product (peel, pulp and seeds) with water, at the temperature of 24 °C, after which they were dried in a tray dryer (Memmert UF30, City, Schwabach, Germany) at 50 °C until its moisture content reached less than 10% (wet basis). This was to limit the loss of available bioactive compounds which are still available even at high levels of drying temperature, namely 60 °C according to Nour et al. [15] or 70 °C according to Sogi et al. [22]. After cooling, the tomato seed were ground in an electrical mill (Heinner, Navy 150, Guangdong City, China). Sieving was made on sieves using a vibratory shaker (Retsch Vibratory Sieve Shaker AS 200 basic, Haan, Germany) in order to obtain particle sizes lower than 500 µm. The raw materials were analyzed according to the international or Romanian standard methods. The moisture content was determined according to International Association for Cereal Chemistry (ICC) method 110/1, ash content according to ICC 104/1, protein content according to ICC 105/2, fat content according to ICC 136. The wheat flour mixes (blends) were analyzed also for the falling number value according to ICC 107/1. The gluten deformation and wet gluten content were analyzed according to the Romanian standard SR 90/2007.

### 2.2. Flour Composites

Flour composites were obtained from the wheat flours which were substituted by tomato seed flour at the levels of 0%, 5%, 10%, 15% and 20%. The wheat–tomato seed composite flours were analyzed for their moisture content according to ICC 110/1, fat content according to ICC 136, protein content according to 105/2, ash content according to ICC 104/1 and falling number according to ICC 107/1. According to previously studies [15] it seems that the incorporation of tomato pomace in wheat flour up to 10% level did not have a negative effect on the acceptability of bread. Also, according to Carlson et al. [34], from the technological point of view, high levels of 20% tomato seed flour addition in wheat flour increased loaf volume of bread. However, higher levels such as 25% tomato pomace addition in wheat flour presented a negative effect on the flavor quality of bakery products as Bhat and Ahsan [30] reported.

### 2.3. Evaluation of Wheat–Tomato Seed Composite Flours on Mixolab Dough Rheological Properties

The Mixolab device (Chopin, Tripette et Renaud, Paris, France) was used to analyze the mixing and pasting behavior of wheat–tomato seed composite flours according to ICC standard method No.173. The tests were made for each sample in order to achieve the optimum dough consistency of 1.1 Nm. The evaluated parameters from the Mixolab curve were the water absorption capacity (WA), dough development time (DT), dough stability (ST), minimum torque value corresponding to the initial heating (C2), maximum torque value corresponding to the heating stage (C3), torque value corresponding to the stability of hot starch paste (C4), torque value corresponding to the final starch paste viscosity after cooling (C5) and the difference between Mixolab torques C1 and C2 (C1-2), C2 and C3 (C3-2), C3 and C4 (C3-4), C4 and C5 (C5-4).

### 2.4. Evaluation of Wheat–Tomato Seed Composite Flours on Rheological Properties

Oscillatory and creep and recovery tests were performed at 25 °C using a HAAKE MARS 40 rheometer (Termo-HAAKE, Karlsruhe, Germany) with non-serrated parallel plate geometry with a diameter of 40 mm and a gap width of 2 mm.

The dough samples were prepared at optimum Farinograph water absorption by mixing until full dough development by using a Brabender Farinograph®-E with a 300 g capacity (Brabender OHG, Duisburg, Germany). The composite flour of 14% moisture basis was kept in the Farinograph bowl and during mixing water was added from the burette to give a dough consistency of 500 BU. Then, the sample prepared was placed between plates and rested for 5 min to allow relaxation and temperature stabilization. The mechanical spectra were measured in a range of linear viscoelasticity, at constant stress of 15 Pa, in frequency sweeps from 1 to 20 Hz. The range of linear viscoelasticity was established based on the dependence of storage modulus (*G’*) and loss modulus (*G”*) on stress in the region 0.01–100 Pa, at constant oscillation frequency of 1 Hz. The experimental data acquired were described by the power model [37,38]:(1)$$G^{\prime }(\omega )=K^{\prime }\cdot \omega^{n}{}^{\prime }$$
(2)$$G^{\prime \prime }(\omega )=K^{\prime \prime }\cdot \omega^{n^{\prime \prime }}$$
where: *G’* is storage modulus (Pa), *G”* is loss modulus (Pa), ω is angular frequency (rad/s), *K’*, *K”* (Pa·s*^n’^*), *n’*, *n”* are experimental constants.

Creep-recovery tests were performed in the range of the linear viscoelasticity at constant stress of 50 Pa. The creep stage time was of 60 s, and the recovery stage was 180 s. The obtaining data of strain as a function of time were expressed in the form of compliance using the following equation [39]:*J(t) = γ(t)/σ*(3)
where *J* (Pa^−1^) is compliance, *γ* is the strain and *σ* is the constant stress applied during the creep test (Pa^−1^).

Experimental data from creep and recovery tests were analyzed by means of a compliance rheological parameter and fitted to the parameter Burger model [40,41] using the Equation (4) for the creep phase and Equation (5) for the recovery phase.
*J(t) = J_Co_ + J_Cm_* (1 − exp*(−t/λ_C_*)) *+ t*/*μ**_Co_*(4)
*J(t)* = *J_max_* − *J_Ro_* − *J_Rm_*(1 − exp(−*t*/*λ**_R_*))(5)
where *J_io_* (Pa^−1^) is the instantaneous compliance, *J_im_* (Pa^−1^) is the retarded elastic compliance or viscoelastic compliance, *t* (s) is the phase time, *λ_i_* (s) is the retardation time, *μ_Co_* (Pa·s) is the zero shear viscosity and *J_max_* (Pa^−1^) is the maximum creep compliance obtained at the end of the creep test. The recovery compliance, *J_r_* (Pa^−1^), evaluated where dough recovery reached equilibrium, is calculated by the sum of *J_Ro_* and *J_Rm_*. The relative elastic part of the maximum creep compliance, expressed as percent recovery was determined using Equation (6) [42,43]:(6)$$Recovery\;(\%)=\frac{J_{r}}{J_{max}}\cdot 100$$
where: *J_max_* is the maximum creep compliance value in the creep phase for the 60 s, which corresponds to the maximum deformation, and *J**_r_* is the compliance value at the end of the recovery phase.

### 2.5. Microstructure of Flour Composite Dough

Epifluorescence light microscopy (EFLM) was used to characterize the microstructure of wheat flour dough with different levels of tomato seed flour addition. Dough microstructure was analyzed using a Motic AE 31 inverted microscope (Motic, Optic Industrial Group, Xiamen, China) operated by catadioptric objectives LWD PH 203 (N.A. 0.4). A thin portion was cut from the dough sample and dipped in a fixing solution composed of 1% rhodamine B and 0.5% fluorescein (FITC) in 2-methoxyethanol obtained from Sigma-Aldrich, Germany for at least 1 h. Fluorescein and rhodamine B was used as two fluorescent dyes specific for detecting starch and proteins in the dough samples. Fluorescein detects starch and rhodamine B proteins from the dough system. The EFLM images were analyzed using ImageJ (v. 1.45, National Institutes of Health, Bethesda, MD, USA) software according to Peighambardoust et al. [44] and Codină and Mironeasa [45,46].

### 2.6. Statistical Analysis

The experimental data and fitting parameters of the employed models were statistically evaluated by one-way analysis of variance (ANOVA) followed by the least significant difference (LSD) test at significance level of 0.05 using SPSS software (trial version, IBM, Armonk, NY, USA). The goodness of fit of the models was assessed using the corresponding determination coefficients (*R*^2^).

## 3. Results and Discussions

### 3.1. Flour Characteristics

The analytical characteristics of wheat flour samples were as following: 8.00 mm gluten deformation index, 58.50–62.90% water absorption, 14.50% moisture, 33.00% wet gluten and 0.65% ash content. According to the wheat flour analytical data these were of a very good quality for bread making [47]. The wheat flour used in our study presented high FN values (445 s) indicating that they had a low α amylase activity.

The tomato seed flour presented the following chemical characteristics (dry basis): 6.94/100g moisture content, 29.50/100g protein content, 19.50/100g fat content and 3.92/100g ash content. According to the data obtained, the tomato seed flour was rich in fat and proteins, these values being in agreement with those reported by Mechmeche et al. [25] and Del Valle et al. [9] for tomato seeds.

### 3.2. Wheat–Tomato Seed Composite Flours Physicochemical Characteristics

The physicochemical characteristics of the wheat–tomato seed composite flours are shown in Table 1. As it may be seen in the composite flours the fat, protein and ash increased with the increase of tomato seed flour addition level whereas the moisture decreased. This was expectable due to the fact that tomato seed flour has a higher amount of fat, proteins and ash and a lower moisture value compared to wheat flour. For the samples in which 20% tomato seed flour was incorporated the values of the fat and ash content were tripled and doubled compared to the control sample whereas the protein content values increased by 27.5%.

The Falling Number (FN) values increased at an addition level up to 10% tomato seed after which its values slightly decreased. This fact indicates that when high levels of TSF were added in wheat flour the blend flour slurry viscosity began to decrease probably due to the decreased viscosity of the starch amount from the samples. It is well known that the flour slurry given by the FN is inversely correlated with α amylase activity [48]. At low levels of TSF addition, the flour mixes viscosity increased and at high levels it slightly decreased, leading to different FN values.

### 3.3. Influence of Tomato Seed Flour on Mixolab Dough Rheological Properties

The addition of tomato seed flour (TSF) to wheat flour dough decreased the water absorption values (Table 2) which were 6.2% lower for the dough sample with 20% TSF addition compared to the control. This decrease of the wheat–tomato seed composite flours water absorption may be attributed to the decrease of the gluten content in the blends, which needs less hydration as a result of TSF incorporation, which is gluten free flour [49]. Also, lipids from TSF may partially coat the starch granules and gluten proteins, decreasing the water absorption during mixing [50]. In the mixing stage, both Mixolab parameters dough development time (DT) and stability (ST) increased for the TSF addition up to 20% and 10% respectively. Contrary, a decrease of the farinograph parameters, dough development and stability was reported by Majzoobi et al. [51]. The increase of DT and ST was significant (*p* < 0.05) and may be attributed to the TSF capacity of foaming and emulsifying properties [5]. This will favor the foaming and emulsifying activity of proteins from the dough system, leading to a more stable three-dimensional network structure [52]. The TSF contains a high amount of lipids which interact with proteins and starch. These interactions are facilitated by TSF emulsifying capacity which can bridge the lipids to the gluten proteins or to the starch granules helping the dough to become more stable and structured [53]. However, at levels higher than 10% TSF addition to wheat flour the ST started to decrease probably due to the gluten dilution from the dough system. Compared to the control sample, the dough weakening was insignificant since for all the samples with TSF addition the ST values were higher.

The Mixolab parameters related to protein weakening (C2 and difference between the points C1 and C2) were higher for the samples in which tomato seed flour was incorporated (Table 2). These results indicated the fact that dough samples with TSF were stronger than the control sample. However at high levels of TSF addition the C2 value slightly decreased compared to the control sample, probably due to the high gluten dilution of the composite flours.

Tomato seed proteins contain high levels of the globulin fraction [54] which during heating might expose hydrophobic groups which can generate some interaction between proteins, leading to their aggregation [55]. Also, during heating TSF proteins are capable of gel formation leading to a more elastic dough with a more compact protein network [56] which can be breakdown less, a fact reflected by the increased C2 values.

The Mixolab parameters (C3, C4, C5 and the difference between the points C2 and C3 (C3-2), C3 and C4 (C3-4), C4 and C5 (C5-4) are significantly related to the starch behavior during the different heating and cooling phases [57]. The Mixolab C3 and C3-2 values are associated with the starch gelatinization process. The decrease of these values with the increase level of TSF addition showed that the gelatinization capacity of the dough samples decreased (Table 2). This may be attributed to lower starch content from the wheat–tomato seeds composite flour samples. As the amount of TSF in wheat flour increased, the non-starch component from the dough system decreased, leading to a decrease of the dough viscosity when temperature increased above the starch gelatinization.

The C4 torque reflects the hot starch stability paste was slightly increased by 2.58% at levels up to 5% TSF (Table 2). The initial increase of C4 was an expectable one since this value is related to the amylolitic activity from the dough system which decreased with the amount of increased TSF addition. This fact leads to an increased dough viscosity and thus to an increase of the C4 value. Also, compared to the control sample, the difference between C3 and C4 value (C3-4) presented lower values for the samples with TSF addition, a fact that reflects a decrease of the starch degradation rate. This, along with the C4 increase, indicates a more stable gel [58]. However, at high levels of TSF addition the C4 value decreased, this value being 9.67% lower than of the control sample when 20% TSF was incorporated in the dough system. These results may be associated to the protein denaturation from the TSF which according to Sarkar et al. [55] are optimum in the temperature range of 80–90 °C. According to previous reports [55] at the temperature of 87 °C the tomato seed protein showed significant denaturation caused by its globular and non-globular protein components. The protein denaturation from TSF will lead to changes in the dough system conformation of which viscosity begins to decrease. Also, by its denaturation the TSF proteins lose their capacity to retain water. This can be retained in a higher amount by the starch granules of which the gelatinization process becomes more complete, a fact that may favor its amylolitic atacability.

For the dough samples in which TSF were added in wheat flour, C5 and the difference between the C5 and C4 peaks (C5-4) presented higher values than for the control (Table 2). This fact indicated that by TSF addition in wheat flour the degree of starch retrogradation increased. However, when high levels of TSF were added, the C5 and C5-4 began to decrease probably due to the lipid content of the TSF which may have interacted with starch and proteins from the dough system, favoring less starch retrogradation [59].

### 3.4. Influence of Tomato Seed Flour on Dynamic Dough Rheological Properties

Figure 1 shows the mechanical spectra of dough samples formulated with TSF at different levels. Both *G’* and *G”* moduli values increased with the increase of TSF level addition in wheat flour. The increase of the dynamic moduli can be due to the limited plasticization effect and to the TSF emulsifying properties [5] and its lipids contents which favor gluten aggregation and gives rise to a more elastic behavior [60]. The TSF proteins are not similar to the gluten proteins and thus by TSF addition the *G”* values also increased indicating a more viscous behavior of the dough samples, probably due to the dilution effect on gluten proteins. A similar trend for the dynamic moduli was also found when wheat flour was substituted with grape seeds flour [42]. However, for all the analyzed samples the *G’/G”* values were higher than 1, indicating a solid elastic-like behavior of dough [61] with and without TSF addition in wheat flour.

The parameters for power-law equations which were used to describe the dependence of moduli on the oscillation frequency are shown in Table 3. As it may be seen, the dependency of *G’* and *G”* dynamic moduli on the oscillation frequency was well modeled by Equations (1,2) in the range of tested frequency from 1 to 20 Hz. The coefficient of determination (*R*^2^) values was higher than 0.999 and 0.960 for *G’* and *G”*, respectively, showing that the power law model was adequate in modeling the viscoelastic properties of dough samples. The obtained values of *K’*, *K”*, *n’* and *n”* parameters adequate to *G’* and *G”* are shown in Table 3.

The values of *K’* and *K”* increased with the increase of TSF addition level in wheat flour. Significant increases (*p* < 0.05) were noticed especially between the control sample and samples in which 15% and 20% TSF were incorporated. The highest values of *K’* and *K”* were obtained for the dough sample with the highest level of TSF addition in wheat flour. The results obtained indicated that the TSF addition led to a stronger dough compared to the control sample. This effect on the viscoelastic properties can be related to the TSF emulsifying properties which act as a filler in the dough viscoelastic matrix [62] causing strong bonds which lead to higher modulus values.

The results obtained for the *G’* slope represented by *n’* presented lower values than *n”* for the *G”* slope (Table 3). The TSF addition in wheat flour dough favors the binding of lipids from TSF to gluten proteins by hydrophobic interactions during mixing. Also, due to the TSF emulsifying capacity the gluten proteins charge decreased, favoring their aggregation, leading to an increase of the *n’* and *n”* parameters values compared to the control sample. The additional level of TSF had a significant effect (*p* < 0.05) on *n’* at level of 15% and 20%, while on *n”* only at the level of 15%. Proximate *n’* and *n”* values (*n’* = 0.22–0.12; *n”* = 0.23–0.181) were found by Chouaibi et al. [63] for wheat flour dough with commercial tomato products addition.

Figure 2 showed creep and recovery curves for wheat flour dough with different levels of TSF addition in wheat flour. The replacement of wheat flour with TSF caused an increase of the deformation compliance only for the sample with 15% TSF addition, as compared to control. The other blends samples showed lower compliance values during the creep test, indicating lower deformability than the control wheat flour dough. The lowest deformation compliance obtained for the sample with 20% TSF addition in wheat flour shows a lower deformability and therefore a stronger matrix for the dough structure.

The experimental data of compliance were well adjusted (*R*^2^ > 0.97) to the Burgers model, Equations (4,5). The parameters of Burgers model are shown in Table 4. In general, the levels of TSF addition exert a significant effect on the creep parameter values. The highest value of instantaneous compliance was observed in the case of the control sample, indicating a more elastic nature and higher recovery. An addition of TSF led to a decreased of *J_Co_*, except for the sample with 15% TSF which showed a slightly increase of the instantaneous compliance. This increase can be related to a slight firmness improvement of dough structure, in according to the result reported by Miś [64]. The highest change on *J_Co_* was found for the sample with 20% TSF, indicating a significantly (*p* < 0.05) decrease of the instantaneous elasticity. In respect to the retardation times obtained for the creep and recovery phase, the results showed that the TSF addition levels exhibit significance changes (*p* < 0.05) on *λ* parameter. The increase of *λ_C_* values for the samples with levels higher than 5% TSF addition in wheat flour is significant, showing that the retarded elastic creep took place more slowly. The decrease of *λ_R_* values with the increase of TSF addition level in wheat flour to 15% and 20% showed that the retarded elastic recovery took place more rapidly. A significant higher viscosity, *μ_Co_* was found in the all samples with TSF addition, except the sample when 15% TSF was incorporated in wheat flour, indicating a higher dough resistance to flow deformability than the control sample. Therefore, the samples with 15% TSF addition in wheat flour presented lower opposition to deformation than the dough samples with 5%, 10% and 20% TSF. This behavior is correlated to the maximum creep compliance value (9.41) obtained for sample with 15% TSF. Compared to other samples with TSF addition, this effect can be related to the gluten dilution in dough. The decrease of gluten proteins by TSF addition at constant stress led to an increase of creep compliance, whereas the elastic contribution decreased [65].

The recovery data obtained (Table 4) shows the elasticity variation of dough samples with TSF addition at different levels to wheat flour. The level of TSF has a significant influence (*p* < 0.05) on the instantaneous recovery compliance and the retardation time, but not on the recovery parameter (*J_r_/J_max_*). The control dough showed the lowest elasticity. The highest elasticity was found in the sample with 20% TSF addition, which could be associated with its higher *J_Ro_* value. A higher compliance elasticity during recovery suggested higher recoverable energy stored by a more cross-linked gluten than the control sample. This fact may be due to the increased aggregation between gluten proteins in the dough system with TSF addition [66]. Among samples with TSF, the highest elasticity was found for the samples in which 15% and 20% TSF were incorporated in wheat flour.

### 3.5. Influence of Tomato Seed on Dough Microstructure

Dough samples with different levels of tomato seed flour addition were analyzed with EFLM (Figure 3A–E). All the samples were labeled with a solution of 1% rhodamine B and 0.5% fluorescein (FITC) in 2-methoxyethanol. In a dough system rhodamine B will label protein in red and FITC will label starch in green. Figure 3A shows a dough sample without tomato seed flour addition. In this sample, the presence of a very high amount of starch granules that were connected to the protein network which was squeezed in the starch matrix was observed. Comparing the structure images obtained for the dough samples with different levels of tomato seed addition a significant difference may be seen between the amount of starch granules and protein. When the level of tomato seed addition increased in wheat flour a less green area was seen and a more red area was present in the dough system (Figure 3B–E). This fact indicated that the starch granules became fewer and the protein content higher. Taking into account that tomato seeds present more than double the amount of proteins compared to wheat flour, it was expectable that the starch content began to decrease and the protein to increase from the dough matrix with the increase of tomato seed level addition. When low levels of tomato seed flour were incorporated in the dough system the starch granules were glued together in the matrix.

At high levels of tomato seed flour addition the starch granules appeared separated surrounded by the protein network. However, all the samples analyzed seemed to be homogeneous, with no black regions in the matrix, meaning that the dough did not become too weak even when higher levels of tomato seeds were incorporated. The results obtained from EFLM were in accordance with the rheological data. Even if the wheat flour substitution level with TSF was of 20% the Mixolab stability of the dough system was higher than the stability of the control samples whereas the both moduli values *G’* and *G”* increased with the level of TSF addition increase indicating that dough with TSF was a stronger and compact one.

## 4. Conclusions

In conclusion, the wheat-tomato seed composite flours parameters analyzed, namely protein, fat and ash, increased with the level of tomato seed addition increase in wheat flour, indicating the fact that bread with TSF will present a higher nutritional and energetic value for consumers. According to the rheological data, all the dough samples with TSF addition were more stable and strengthened compared to the control sample presenting higher values for dough stability and development time. This fact indicates to the bakers that the dough with TSF addition presents a higher capacity to keep its shape during proofing and therefore, it may be fermented for a longer period of time. Also, dough with TSF addition will present a higher ability to retain the gas formed during the fermentation process, a fact that will influence bread porosity. The higher dough strength for the samples with TSF indicated that the bread obtained presented a more firm texture. Regarding the mixing behavior for the dough samples with TSF addition, it seemed that according to the Mixolab data the bakers should mix the composite flour for a longer period of time and must introduce less water amount in order to obtain an optimum dough consistency. At high levels of TSF addition in wheat flour, all the Mixolab parameters torques related to the starch behavior (C3, C4, C5) decreased probably due to the decrease of the starch content from the wheat–tomato seed flour samples. At low levels of addition, TSF seemed to integrate very well into the dough system increasing its starch stability paste. Also, the FN increased when low levels of TSF were incorporated and decreased when high levels of TSF were added in dough samples. This, along with starch behavior recorded with the Mixolab device, showed that the addition of α amylase may be recommended for wheat flour with high FN values without affecting in a negatively way the dough stability, a fact that will lead to good bakery products quality. It must been mention that when high levels of TSF were added the C5 and C5-4 begin to decrease, favoring less starch retrogradation, indicating the fact that bakery products obtained with TSF addition will present a higher shelf life than the control sample. The dynamic rheological data showed that with the increase of TSF addition level in wheat flour the dough samples presented higher viscoelastic solid properties. Creep–recovery tests showed that TSF addition in wheat flour led to dough with higher resistance to deformation, smaller creep strain values and higher recovery, indicating stronger dough with greater elasticity. This indicated that during the technological process the dough samples with TSF addition presented the capability to maintain their form, a fact that will create the possibility to prolong the fermentation phase due to a higher ability to retain the gas formed during this process. Also, after baking the bakery products may present higher loaf volumes and porosity due to the fact that by prolonging the fermentation process the amount of the gas released will increase and may be retained by the dough system due to it high elasticity. The results obtained from EFLM were in accordance with the rheological data. All the analyzed samples were homogeneous, with a compact dough matrix, showing that the dough samples were strong even when high levels of TSF were incorporated. According to the data obtained, we concluded that tomato seed, as a whole flour form, can be used in bread making by blending with wheat flour even at high levels of up to 20%, as an alternative for developing new bakery products. The TSF addition increased the dough strength which makes us recommend its use for wheat flour of a good quality for bread making with a high content of wet gluten. In the case of its addition in wheat flour with high FN values we recommend the addition of α amylase in the dough system, a fact that will increase the amount of the gas formed in the dough system, gas that dough with TSF addition is capable of retaining.

## Figures and Tables

**Figure 1 foods-08-00626-f001:**
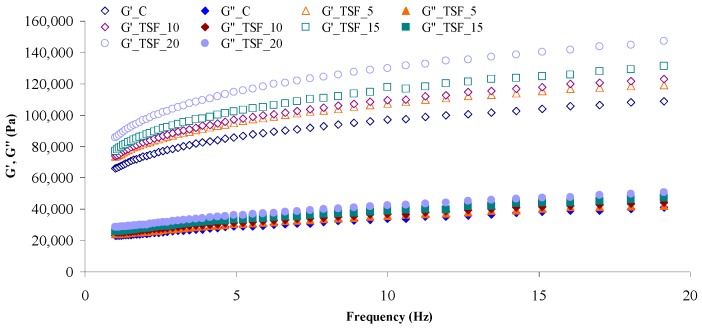
Mechanical spectra of control dough (C) and dough samples with different levels (5%, 10%, 15% and 20%) of tomato seeds flour (TSF). Presented data are mean values.

**Figure 2 foods-08-00626-f002:**
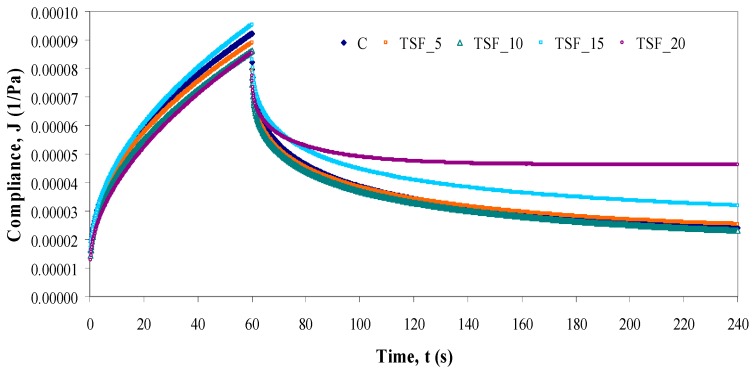
Creep and recovery curves of control sample (C) and dough samples with different levels (5%, 10%, 15% and 20%) of tomato seed flour (TSF). Presented data are mean values.

**Figure 3 foods-08-00626-f003:**
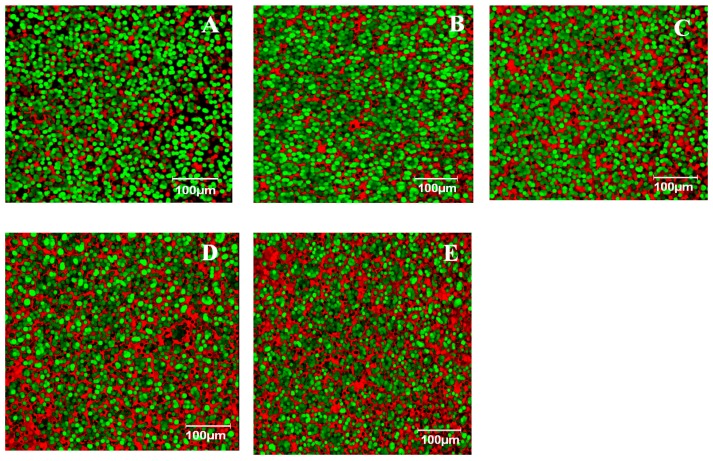
Microstructure taken by epifluorescence light microscopy (EFLM) of wheat dough with tomato seed flour (TSF) at different levels: 0% (**A**), 5% (**B**), 10% (**C**), 15% (**D**) and 20% (**E**). Green, starch granules; red, protein.

**Table 1 foods-08-00626-t001:** Physicochemical characteristics for the wheat–tomato seed composite flours.

Sample	Protein (%)	Lipids (%)	Ash (%)	Moisture (%)	Falling Number (s)
Control	12.40 a (0.20)	1.60 a (0.10)	0.65 a (0.01)	14.50 a (0.30)	445.00 a (14.00)
TSF_5	13.25 b (0.19)	2.49 b (0.09)	0.80 a (0.00)	13.83 a (0.29)	467.00 ab (14.00)
TSF_10	14.11 c (0.18)	3.39 c (0.09)	0.97 a (0.00)	13.47 d (0.27)	478.00 b (12.00)
TSF_15	14.96 d (0.17)	4.28 d (0.09)	1.13 a (0.00)	13.11 c (0.25)	445.00 a (15.00)
TSF_20	15.82 e (0.16)	5.18 e (0.08)	1.30 b (0.00)	12.74 b (0.24)	434.00 ac (16.00)

Values in parentheses are standard deviations. ^abcde^ Values with the same letter are not significantly different according to the least significant difference (LSD) test (*p* < 0.05).

**Table 2 foods-08-00626-t002:** Water absorption and Mixolab parameters of tomato seed–wheat flour blends.

Sample	WA (%)	ST (min)	DT (min)	C2 (N-m)	C1-2 (N-m)	C3 (N-m)	C3-2 (N-m)	C4 (N-m)	C3-4 (N-m)	C5 (N-m)	C5-4 (N-m)
Control	60.70 a (2.20)	8.53 a (0.01)	3.85 a (0.62)	0.49 a (0.02)	0.60 a (0.02)	1.82 a (0.07)	1.33 a (0.06)	1.55 a (0.09)	0.27 a (0.02)	2.36 a (0.06)	0.81 a (0.03)
TSF_5	59.20 e (0.02)	8.73 e (0.02)	3.98 d (0.03)	0.54 b (0.03)	0.64 b (0.03)	1.73 d (0.06)	1.19 d (0.03)	1.59 d (0.07)	0.14 c (0.02)	2.81 b (0.05)	1.22 b (0.02)
TSF_10	58.50 d (2.20)	9.18 b (0.02)	4.05 c (0.05)	0.55 b (0.03)	0.62 c (0.01)	1.67 c (0.02)	1.12 c (0.01)	1.57 d (0.04)	0.10 b (0.02)	2.78 b (0.05)	1.21 b (0.01)
TSF_15	57.60 c (2.20)	9.08 c (0.01)	4.32 b (0.17)	0.53 c (0.02)	0.64 b (0.02)	1.52 b (0.05)	0.99 b (0.03)	1.35 b (0.03)	0.17 c (0.02)	2.57 c (0.05)	1.22 b (0.02)
TSF_20	56.80 b (2.20)	8.90 d (0.10)	4.55 b (0.01)	0.53 c (0.02)	0.65 b (0.01)	1.50 b (0.03)	0.97 b (0.01)	1.40 c (0.01)	0.10 b (0.02)	2.53 d (0.04)	1.13 c (0.03)

WA, water absorption; Mixolab parameters: ST, stability; DT, development time; C3, C5, maximum consistency during stage 3, stage 5; C2, C4, minimum consistency during stage 2, stage 4; C1-2, difference of the points C1 and C2; C3-2, difference of the points C3 and C2; C3-4, difference of the points C4 and C3; C5-4, difference of the points C5 and C4. Values in parentheses are standard deviations. ^a,b,c,d,e^ Values with the same letter are not significantly different according to the LSD test (*p* < 0.05).

**Table 3 foods-08-00626-t003:** Parameters of power law models, Equations (1,2) describing the dependence of storage (*G*’) and loss (*G*”) moduli on the frequency.

Sample	*G’* = *K’·**ω^n”^*		*G”* = *K”·**ω^n”^*	
	*K’* (Pa s*^n^*^’^)	*n’*	*K”* (Pa s*^n^*^”^)	*n”*
Control	65,707.45 a (7373.07)	0.168 a (0.008)	21,489.72 a (3403.12)	0.196 a (0.008)
TSF_5	73,158.88 a (3875.74)	0.167 a (0.001)	22,721.65 a (439.82)	0.199 a (0.001)
TSF_10	73,711.68 a (2893.63)	0.172 a (0.004)	23,469.96 a (2119.38)	0.197 a (0.005)
TSF_15	77,350.93 b (2412.14)	0.177 c (0.002)	24,470.72 a (1762.34)	0.205 b (0.003)
TSF_20	85,702.71 c (7512.53)	0.182 b (0.002)	26,867.43 b (2610.38)	0.198 a (0.004)

Values in parentheses are standard deviations. ^a,b,c^ Values with the same letter are not significantly different according to the LSD test (*p* < 0.05).

**Table 4 foods-08-00626-t004:** Parameters of Burger’s model.

Sample	Creep Phase					Recovery Phase				
	*J_Co_*⋅ 10^5^ (Pa^−1^)	*J_Cm_*⋅ 10^5^ (Pa^−1^)	*λ_C_* (s)	*μ_Co_*⋅ 10^−6^ (Pa ⋅ s)	*J_max_*⋅ 10^5^ (Pa^-1^)	*J_Ro_*⋅ 10^5^ (Pa^−1^)	*J_Rm_*⋅ 10^5^ (Pa^−1^)	*λ_R_* (s)	*J_r_*⋅ 10^5^ (Pa^−1^)	*J_r_/J_max_* (%)
Control	2.50 a (0.16)	8.13 a (0.04)	36.97 a (0.48)	1.48 a (0.01)	9.13 a (0.33)	2.30 a (0.20)	3.90 a (0.29)	38.19 a (3.47)	6.20 a (0.10)	67.98 a (3.50)
TSF_5	2.45 b (0.03)	7.80 b (1.07)	36.60 b (2.93)	1.60 b (0.24)	8.80 d (0.61)	2.62 b (1.28)	3.61 b (0.45)	37.10b (9.63)	6.22 b (0.83)	70.37 a (4.55)
TSF_10	2.32 e (0.12)	7.69 c (0.64)	37.95 c (0.21)	1.54 c (0.14)	8.51 a (0.54)	2.42 e (0.16)	3.60 a (0.54)	39.11 c (4.74)	6.01 a (0.39)	70.66 a (0.14)
TSF_15	2.51 c (0.36)	8.64 d (1.24)	38.56 e (0.96)	1.45 e (0.22)	9.41 b (1.32)	3.32 c (0.28)	3.49 a (0.74)	36.02d (3.66)	6.81 c (1.02)	72.27 a (0.70)
TSF_20	2.04 d (0.15)	8.41 e (1.60)	42.54 d (3.33)	1.54 d (0.26)	8.48 c (1.27)	4.51 d (2.04)	2.32 c (0.45)	22.48 e (17.21)	6.83 d (1.59)	79.56 a (6.89)

Values in parentheses are standard deviations. ^abcde^ Values with the same letter are not significantly different according to the LSD test (*p* < 0.05).

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
