# Peer review of "Dough Rheological Behavior and Microstructure Characterization of Composite Dough with Wheat and Tomato Seed Flours"

_foods, 2019, doi:10.3390/foods8120626_

Round 1

Reviewer 1 Report

The manuscript foods-639477-peer-review-v1: “Dough rheological Behavior and Microstructure Characterization of Composite Dough with Wheat and Tomato Seed Flours” investigates the effects of added tomato seed flours on the microstructure and dough rheological behaviour.

The manuscript lacks novelty. The literature gives very good samples of investigation of the incorporation of tomato seed flours in breadmaking. I consider the subject, and the conclusions that this manuscript presents already reported from the literature. It is not a new subject that need further investigation in the present form. The following articles study the effect of added tomato seed in dough rheology, and bread quality covering even the concentration range used in the present work:

Sogi, D. S., Sidhu, J. S., Arora, M. S., Garg, S. K. & Bawa, A. S. (2002). Effect of tomato seed meal supplementation on the dough and bread characteristics of wheat (PBW 343) flour. International Journal of Food Properties, 5, 563-571. Carlson, B. L., Knorr, D. & Watkins, T. R. (1981). Influence of Tomato Seed Addition on the Quality of Wheat Flour Breads. Journal of Food Science, 46, 1029-1031. Adekoyeni, O. O., Adegoke, A. F. & Ayano, A. E. (2018). Nutritional, functional, sensory and microbial qualities of wheat-tomato seed flour bread. Carpathian Journal of Food Science and Technology, 10, 47-56.

The title states that “Composite Dough with Wheat and Tomato Seed Flours”. There is used one type of flour and one mixture of tomato seed flour. The authors did not report the use of different types of flours (that differ in their breadmaking ability), or investigate the effect of other factors than the amount of added seed flour (possibly optimization), missing to provide something different and useful that is not reported in the literature. Beside the fact that the conclusions give no valuable information to the reader, this section is long and confusing.

Reviewer 2 Report

Dear Editor and Authors,

I admit that all changes have been done as suggested and manuscript can be published in this form.

Regards

Reviewer 3 Report

The work is very interesting and the manuscript is well written.
Some jargon-type descriptions should be re-written and English needs carefully reviewing.
Subchapters 2.1. and 2.2. need minor re-writing due to unprecise description eg. sieving is made on sieves using vibratory shaker not on shaker using sieves....
The mention methods of analysis are international standards and there is only one Romanian method in chapter 2.1. so it is better to re-edit two last phrases.

What type of plates were used in the rheometer? Serrated or not and what diameter?

Round 2

Reviewer 1 Report

The manuscript foods-639477 “Dough rheological Behavior and Microstructure Characterization of Composite Dough with Wheat and Tomato Seed Flours” investigates the effects of added tomato seed flours on the microstructure and dough rheological behaviour. I still believe that the manuscript lacks novelty. The information about the rheological behaviour of the dough at the concentration range used is not new and interesting since the literature gives the needed information linking it with the bread quality. Moreover, the authors are not clear and concise in reporting the main conclusions of the work.